# Effects of MEH-PPV Molecular Ordering in the Emitting Layer on the Luminescence Efficiency of Organic Light-Emitting Diodes

**DOI:** 10.3390/molecules26092512

**Published:** 2021-04-25

**Authors:** Seok Je Lee, Jun Li, Seung Il Lee, Chang-Bum Moon, Woo Young Kim, Jin Cao, Chul Gyu Jhun

**Affiliations:** 1Division of Electronic and Display Engineering, Hoseo University, Asan 31499, Korea; seokjelee7@gmail.com (S.J.L.); ilseung5758@gmail.com (S.I.L.); wykim@hoseo.edu (W.Y.K.); 2Key Laboratory of Advanced Display and System Applications, Shanghai University, Ministry of Education, Shanghai 200072, China; junli2021@163.com; 3Center for Exotic Nuclear Studies, Institute for Basic Science, Daejeon 34126, Korea; cbmoon@ibs.re.kr

**Keywords:** organic light-emitting diode (OLED), molecular crystal, molecular ordering, enhanced conductivity, enhanced efficiency

## Abstract

We investigated the effects of molecular ordering on the electro-optical characteristics of organic light-emitting diodes (OLEDs) with an emission layer (EML) of poly[2-methoxy-5-(2-ethylhexyloxy)-1,4-phenylenevinylene] (MEH-PPV). The EML was fabricated by a solution process which can make molecules ordered. The performance of the OLED devices with the molecular ordering method was compared to that obtained through fabrication by a conventional spin coating method. The turn-on voltage and the luminance of the conventional OLEDs were 5 V and 34.75 cd/m^2^, whereas those of the proposed OLEDs were 4.5 V and 120.3 cd/m^2^, respectively. The underlying mechanism of the higher efficiency with ordered molecules was observed by analyzing the properties of the EML layer using AFM, SE, XRD, and an LCR meter. We confirmed that the electrical properties of the organic thin film can be improved by controlling the molecular ordering of the EML, which plays an important role in the electrical characteristics of the OLED.

## 1. Introduction

The research on organic light-emitting diodes (OLEDs) has propelled the evolution of next-generation display technologies [1,2,3]. The development of brilliant and innovative technologies originated 50 years ago. In 1965, Helfrich and Schneider discovered the electroluminescence (EL) phenomenon using the organic compound anthracene crystal [4]. This discovery attracted much attention and inspired the development of OLEDs. However, the efficiency was extremely low, and the driving voltage was too high.

In 1987, Tang et al. dramatically reduced the operating voltage by proposing multilayer thin-film light-emitting devices using tris-(8-hydroxyquinoline) aluminum (Alq_3_) and diamine [5]. They called the devices organic electroluminescent diodes. These diodes have a thin-film light-emitting element of organic material and can be applied to display devices to provide visual information with low operating voltage and high efficiency. OLED applications require high luminous efficiency, long lifetime, high color purity, and simple fabrication processes. To accomplish these requirements, many studies have been carried out on a number of methods, such as adding dopants [6], exploiting the phosphorescence concept in the light-emitting layer [7,8,9,10], fabricating stacked structures [11], adding nanoparticles [12], fabricating tandem structures [13], exploiting the thermally activated delayed fluorescence (TADF) phenomenon [14].

From the manufacturing point of the fabrication processes can be classified into the evaporation process and the solution process. The evaporation process is widely used for small molecular materials. The organic material is evaporated in a vacuum to achieve vapor deposition. On the other hand, solution process is normally used for polymer materials. The polymer is dissolved in an organic solvent, and the solution is used for thin film coating by various coating methods (spin-coating, ink-jet printing, aerosol-jet, screen-printing, stamping, blading, template-guided solution-shearing, and so on) [15,16,17]. The solvent is then removed to form an organic thin film.

Burroughes et al. first proposed the solution process using poly(p-phenylene vinylene) in 1990 [18]. Recently, Chang et al. reviewed the comprehensive study on the solution process from polymer solution treatments by tuning the solvent solubility to the film deposition techniques, which affect the device performance [17,19,20,21,22,23,24,25,26,27,28]. Heat treatment plays an important role in improving the order of polymeric films [29,30].

Since the manufacturing process using solution process is simple, low-cost, and better for large-area deposition, so many studies have focused on them for polymer OLEDs [31,32]. To optimize the solution process, it is necessary to find proper solvents for the organic material and suitable coating methods to increase the efficiency of OLEDs. Subsequent studies have been done on the fabrication of multi-layered structures [33,34,35] and applications for polymers with low molecular weight [36,37]. Various coating methods for solution process has also been applied to improve the efficiency of OLEDs [24,38,39]. There has been no research on the correlation between the coating method of the solution process and the device performance OLED until now.

In this study, an emission layer (EML) of poly[2-methoxy-5-(2-ethylhexyloxy)-1,4-phenylenevinylene] (MEH-PPV) in the OLED was fabricated by a conventional spin-coating method and a simple wire-bar-coating process proposed by Khim [24]. We investigated the effects of molecular ordering induced by two different methods on the electrical and optical characteristics of the OLED. We verified the molecular ordering due to different solution process methods, and the device performance were compared. The effect of molecular ordering on the luminescence efficiency and electrical conductivity was also confirmed by analyses with various methods.

## 2. Materials and Fabrication Methods

Figure 1 shows our device structure and the energy level diagram of the device. To clearly verify the effects of molecular alignment on the electrical and optical characteristics of the OLED device, it was fabricated with the simple structure. PEDOT:PSS and MEH-PPV were used for the hole transport layer (HTL) and EML, respectively. The LUMO and HOMO energy levels of MEH-PPV, which is provided by the material supplier [40], are −3.0 and −5.3 eV, respectively. The LUMO and HOMO energy level of PEDOT:PSS, which is reported in Zhou’s work with the same material [41], are −2.4 and −5.2 eV, respectively.

To fabricate the OLED devices, patterned ITO substrates (sheet resistance of 15 ohm/sq, active layer of 3 × 3 mm) were cleaned by an ultrasonic method with acetone, isopropyl alcohol, and distilled water for 30 min each, followed by UV treatment for 20 min. A HTL of PEDOT:PSS (P. VP AI 4083 Heraeus/Clevios, Leverkusen, Germany) was coated onto the cleaned ITO substrates by a spin coater with a speed of 5000 rpm for 1 min. And then baked at 150 °C for 15 min.

Next, the MEH-PPV (Sigma Aldrich/average Mn 70,000–100,000, St. Louis, MO, USA) dissolved in an organic solvent was coated onto the HTL by two kinds of coating methods to form the EML. MEH-PPV is one of the well-known materials for the solution process [42,43,44,45,46]. MEH-PPV was dissolved into the organic solvent of toluene with a concentration of 0.5 wt%. Mixtures of MEH-PPV and organic solvent were previously prepared by stirring for 48 h at room temperature in a glove box under N_2_ atmosphere.

The two different kinds of fabrication processes are shown schematically in Figure 2. The spin coating was done by 2000 rpm for 1 min. Then, the solvent was dried in a vacuum for 12 h. However, at a constant speed of 2.5 cm/s, the wire-bar-coating process was carried out with a groove bar whose pitch is about 360 μm (OSP-30, S60 from Japanese Genuine, Guangdong, China). Then, the film was dried in a vacuum for 12 h. For both two deposition methods, the coated layers were dried in a vacuum to exclude the heat treatment effect. Figure 2c demonstrates the polymer alignment mechanism by the wire-bar-coating method. The alignment of polymer results from the anisotropy of the rheological properties in organic solution [47]. Moving the wire-bar induced the strong fluid flow in the x direction due to the viscous interaction. The large gradient of the flow velocity near the boundary layers induced a viscous torque on the molecules [48], since the shear stress is coupled with the torque by the viscosity coefficients [49,50]. Molecules near the interface would first be adsorbed on the surface and other molecules in the bulk would be aligned with the molecules pinning on the surface due to the van der Waals force among molecules. For this reason, we named the wire-bar-coating method as the molecular ordering method.

An Al cathode layer was fabricated with a thickness of 100 nm under high vacuum conditions of 5.0 × 10^−7^ Torr using evaporation equipment (Thermal Evaporator System, Daedong High Technologies, Gimpo, Korea). Finally, the device was encapsulated using encap glass in a glove box.

## 3. Results and Discussion

We compared the performance of the OLEDs fabricated using these two coating methods. The molecular ordering of anisotropic molecules is influenced by the flow direction of the fluid as mentioned before [47,48,49,50]. In the case of spin coating, the molecules are arranged in the radial direction because the rotational coating process produces fluid flow in the radial direction. However, with the proposed molecular ordering coating method, the molecules are arranged in the one direction because the molecules are pushed in one direction when they are coated. These methods induce different orderings of the MEH-PPV polymer chains. We analyzed the relationship between the molecular orientation and the performance using various analytical instruments. To verify the effects of the fabrication processes on the electrical properties, the MEH-PPV layers in the OLED device were fabricated by the spin coating and molecular ordering coating methods. With these conditions, the same thickness of emissive layer was fabricated. The fabrication process of other layers, such as PEDOT:PSS layer and Al layer, was done under same conditions mentioned above.

The two different solution processes were optimized to obtain the same thickness of the MEH-PPV layer. To confirm that the same thickness was obtained, we measured the thickness of the MEH-PPV layers fabricated by the spin coating and the molecular ordering coating methods using a gallium focused ion beam-scanning electron microscope (Ga-FIB, TESCAN/LYRA3, Brno-Kohoutovice, Czech Republic). The MEH-PPV layers fabricated by two different methods, which are sandwiched between two Al layers, were observed by Ga-FIB. Figure 3 shows the cross-sectional view of the Ga-FIB images. The thickness of the emissive layer was measured at two points. With the spin coating method, the thicknesses at both points were 41 nm, as shown in Figure 3a. However, for the device fabricated by the molecular ordering coating method, the thicknesses were measured as 41 nm and 46 nm, which may have been caused by the roughness of the base aluminum layer on the substrate.

It was hypothesized that high electrical conductivity can be obtained with ordered molecules resulting from the molecular ordering coating method. To support this hypothesis, we analyzed the device characteristics by atomic force measurement (AFM, EM4SYS/Personal AFM NX II, Gwangju, Korea), spectroscopic ellipsometry (SE, ELLIPSO TECHNOLOGY/Elli-SE, Suwon, Korea), X-ray diffraction (XRD, Lab X/XRD-6100, Kyoto, Japan), and an LCR meter (KEYSIGHT/4285A, Santa Rosa, CA, USA). When we prepared the samples to analyze the characteristics of devices using various equipment, to maintain the same conditions, PEDOT:PSS was first coated onto each substrate. MEH-PPV was coated onto the PEDOT:PSS layer by two different methods. Then the characteristics of the devices were evaluated.

AFM images of the coated surfaces were compared to investigate the surface morphology and the texture fabricated by the different coating methods. Figure 4 shows the AFM images of the emissive layers fabricated by the spin coating method and the molecular ordering coating method. More uniform surface morphology was obtained by the spin coating process than the molecular ordering coating process because the scales of the *z*-axis are 26 nm and 33 nm for the spin coting and the molecular ordering coating, respectively. This is consistent with the Ga-FIB images, in which a uniform thickness of the emissive layer was obtained by the spin coating process. A periodic pattern was observed for the molecular ordering coating conditions, as shown in Figure 4b, which is related to the molecular ordering.

To find the correlation between the periodic patterns and molecular ordering, the dielectric constants in the two orthogonal horizontal directions were measured using SE. The dielectric anisotropy was also compared from the measured dielectric constant values obtained by the different coating methods. we also compared the imaginary value of the dielectric constant ($\varepsilon_{2}$) for the emissive layer with respect to the frequency by SE. This value is associated with the electrical conductivity of the emissive layer as follows [51,52]:(1)$$\hat{\varepsilon }=\varepsilon_{1}+{i\varepsilon }_{2}=\varepsilon_{1}\left(1-i\frac{\sigma }{{\omega \varepsilon }_{1}}\right)=\varepsilon_{1}-i\frac{\sigma }{\omega }$$
where $\hat{\varepsilon }$ is the complex dielectric constant, $\varepsilon_{1}$ is the real value of the dielectric constant, σ is the conductivity, and ω is the angular frequency. In Equation (1), $\frac{\sigma }{\omega }$ can be expressed as the imaginary part of the dielectric constant, which we investigated to find the dielectric anisotropy.

For the SE measurements, the MEH-PPV layers were coated onto PEDOT:PSS on the silicon wafers by two different coating processes. The measured data were analyzed by the Lorenz model. The measured dielectric anisotropy is shown in Figure 5 and can be represented by the orientation order parameter S [53]:(2)$$S=\frac{\varepsilon_{e}-\varepsilon_{0}}{\varepsilon_{e}+2\varepsilon_{0}}$$

We set the order parameter value by the ordering direction of the molecular as the x-direction (extraordinary) and the perpendicular to the ordering direction of the molecular as the y-direction (ordinary). Figure 5 shows measured dielectric constants of MEH-PPV layer by SE. Isotropic characteristics of the imaginary dielectric constant were obtained for the emissive layer made by the spin coating process. However, with the molecular ordering coating process, dielectric anisotropy was obviously obtained, as shown in Figure 5b. This is strong evidence of homogeneous ordering for the anisotropic shape of the polymer molecules.

To analyze the degree of crystallization of the MEH-PPV film, X-ray diffraction measurements (XRD) were performed at 2° to 50° at a rate of 2°/min. Since Bragg diffraction depends on the periodic structure or on degree of alignment, it is useful to evaluate the arrangement of the MEH-PPV main chain. Figure 6 shows the measured XRD data. For the XRD data of MEH-PPV layer fabricated by the spin coating method, there was no specific peak. In other words, no periodic structure was obtained. Therefore, the organic layer fabricated by the spin coating method is an amorphous molecular film. However, from the MEH-PPV layer by the molecular ordering method, the specific peak was obtained at 2θ = 24.9°, which corresponds to the periodic structure of 3.75 Å. This peak correspond to the periodic structure of the main chains of MEH-PPV, which is consistent with the experimental results of Yang’s group [54]. It is confirmed that the molecular crystals phase of the organic layer by the molecular ordering coating method was obtained.

Because the molecular arrangement is strongly related to the conductivity of the organic layer, the LCR measurements were used to make comparison between the electrical conductivity caused by the molecular ordering and amorphous polymer. The electrical characteristics of the devices were analyzed by comparing the conductivity with respect to the frequency. We compared the conductivity of the MEH-PPV layer fabricated by the two types of methods as shown in Figure 7. In this case of molecular ordering coating method, the maximum conductivity of the MEH-PPV layer was 4 × 10^5^ S/m for the molecular ordering coating method and 3 × 10^5^ S/m for the spin coating process. The molecular crystal obtained by the molecular ordering coating method has higher conductivity than the amorphous polymer obtained by the spin coating method. The AFM, SE, and XRD results show a strong correlation between the molecular ordering and conductivity. From the disorder-controlled transport mechanism, the charge mobility is coupled with the positional disorder and the width of the density of state distribution which is also related with polymer alignment [55,56,57,58]. Thus, the electrical characteristics can be improved by controlling the molecular ordering [59,60].

Finally, for OLEDs with the molecular crystal and amorphous polymer, the electro-optical characteristics such as the current density, luminance, luminous efficiency, power efficiency, quantum efficiency and EL spectrum were compared by the I-V-L system (LMS/PR-670, Anyang, Korea). Figure 8 shows the measured I-V-L characteristics of the OLED devices with amorphous polymer applied by spin coating. The turn-on voltage was 5 V, and the maximum luminance was 34.75 cd/m^2^. On the other hand, in the case of the OLED with the molecular crystal layer, the turn-on voltage was 4.5 V, and the maximum luminance was about 120.3 cd/m^2^. The measured electro-optical characteristics of OLED device fabricated by amorphous polymer and molecular crystal were listed on Table 1. The luminous efficiency of the OLED with the molecular crystal and amorphous polymer were 0.123 cd/A and 0.024 cd/A, respectively. The power efficiency of the OLED with the molecular crystal and amorphous polymer were 0.068 lm/W and 0.013 lm/W, respectively. The quantum efficiency of the OLED with the molecular crystal and the amorphous polymer were 0.066 and 0.012%, respectively. Luminous efficiency, power efficiency, quantum efficiency of the molecular crystal are 5 times higher than those of the amorphous polymer. From the EL data of molecular crystal, the shoulder pick was obtained, which is the proof of the alignment of the molecules [61]. The low EQE is due to the simple structure to clearly verify the effects of molecular alignment on the electrical and optical characteristics of the OLED device. Not only EQE but also other efficiencies can be improved further by applying a multi-layer structure with functional layers.

We measured the voltage-current curves of the OLEDs fabricated by the two methods. The electric conductivity (g) can be obtained from the voltage-current curves and is related to the electric charge injection and transfer in the device [55,62]. After the voltage-current curves were measured by the I-V-L system, the electrical conductivity can be obtained from the measured data by a curve fitting method:(3)$$J=g\left(V\right)\;V=g_{l}V^{l+1}$$
where $g\left(V\right)=g_{l}V^{l+1}$ is the effective conductivity per unit area with dimensions of S/m, and l is a dimensionless quantity that describes the property of trap states, Equation (3) expresses the relationship between the current density and the voltage.

At l=0, the device follows Ohm’s law. At l=1, space charge limited current (SCLC) occurs without trap conditions. At l>1, the behavior follows the SCLC of the trapped state. Since both the amorphous polymer and molecular crystal have l>1, they follow the SCLC behavior of the trapped state when charge is transferred in the polymer film.

When charge is injected from the electrode into the organic polymer, the injected charges are transferred by a hopping mechanism. In the case of the organic polymer, the density of state (DOS) has a Gaussian distribution [55]. The wider distribution of the Gaussian DOS, the lower the injection barrier is. The Gaussian DOS of the amorphous polymer is wider than that of the molecular crystal, so the injection energy of the amorphous polymer is lower than that of the molecular crystal. This is consistent with Figure 8, which is presented on a log scale.

In the amorphous polymer, the charge injection is easier than in the molecular crystal. However, as the voltage increases, the current densities converge to a similar level. In addition, excellent luminescence properties of the molecular crystal were obtained. These results show that the Gaussian DOS with the amorphous polymer is more widely distributed, and the energy states are also extensively dispersed, so the mobility of the amorphous polymer is lower than that of the molecular crystal. Thus, with the Gaussian DOS in the ordered organic material, the charges can be easily transferred and form excitons, even if less charge is injected into the organic molecules. Therefore, the electro-optical properties of the OLED device are improved by allowing it to emit more light.

## 4. Conclusions

We have investigated the effects of molecular ordering of MEH-PPV film as EML on the electro-optical characteristics of OLEDs. To clearly verify the effects of molecular alignment on the electrical and optical characteristics of the OLED device, it was fabricated with a simple structure. The EML was fabricated by the conventional spin coating method and the molecular ordering coating method of solution processes. The performance values of the OLEDs were compared. To understand the mechanism of the improved performance of our molecular ordering coating method, we analyzed the properties of the morphologically, dielectric anisotropy, degree of crystallization and conductivity by using AFM, SE, XRD, and LCR meter, respectively. We have confirmed that the electrical properties of the organic thin film can be improved by controlling the molecular ordering in the EML.

The turn-on voltage and the luminescence of the OLED fabricated with the amorphous polymer were 5 V and 34.75 cd/m^2^, while those obtained by the molecular crystal were 4.5 V and 120.3 cd/m^2^, respectively. Luminous efficiency, power efficiency, quantum efficiency of the molecular crystal are five times higher than those of the amorphous polymer. The molecular ordering plays an important role in the electrical characteristics of the OLEDs. The maximum luminance and efficiency can be improved further by applying a multi-layer structure with functional layers. If molecules of organic layer can be aligned, not only the electrical properties of the device can be improved, but also optical properties can be controlled, which enhance the out-coupling efficiency and emit polarized light.

## Figures and Tables

**Figure 1 molecules-26-02512-f001:**
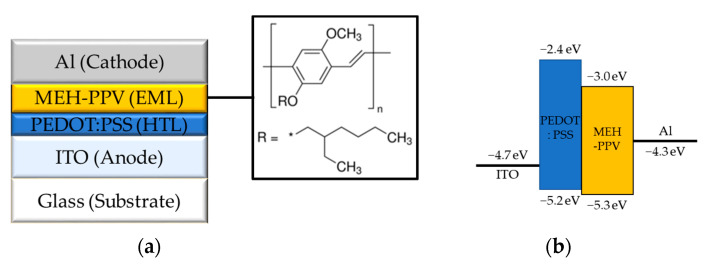
Fabricated OLED with MEH-PPV: (**a**) device structure and (**b**) energy level diagram.

**Figure 2 molecules-26-02512-f002:**
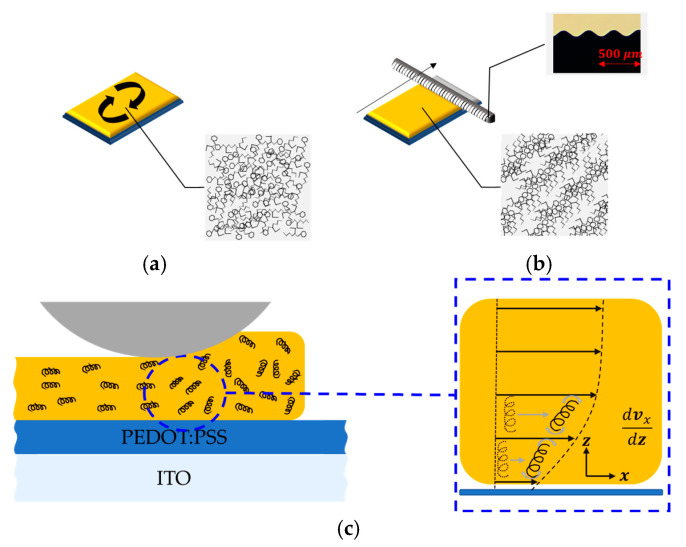
Two kinds of coating methods of MEH-PPV film: (**a**) spin coating and (**b**) molecular ordering coating methods, (**c**) mechanism of molecular alignment by the molecular ordering coating method.

**Figure 3 molecules-26-02512-f003:**
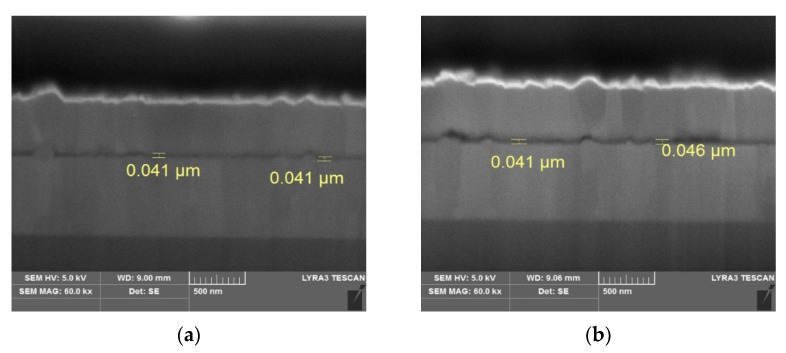
Ga-FIB-SEM images of MEH-PPV film obtained by (**a**) spin coating and (**b**) molecular ordering coating methods.

**Figure 4 molecules-26-02512-f004:**
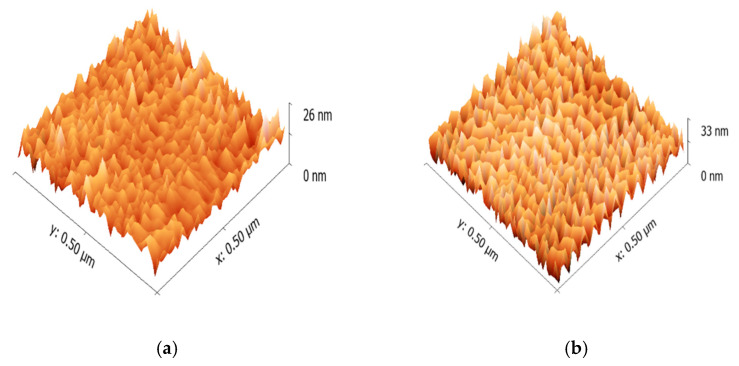
AFM images of MEH-PPV film obtained by (**a**) spin coating and (**b**) molecular ordering coating methods.

**Figure 5 molecules-26-02512-f005:**
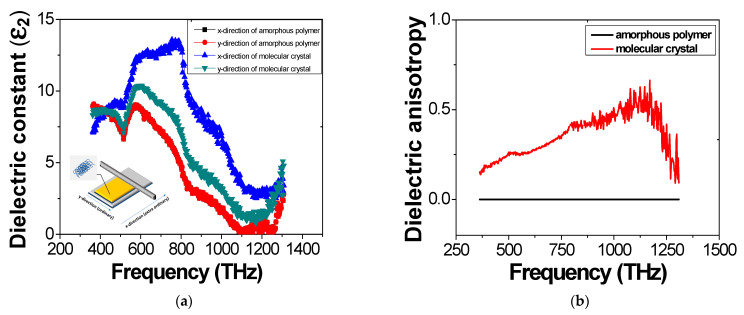
Measured dielectric constants of MEH-PPV film by SE: (**a**) imaginary value of dielectric constants with respect to the frequency and (**b**) dielectric anisotropy.

**Figure 6 molecules-26-02512-f006:**
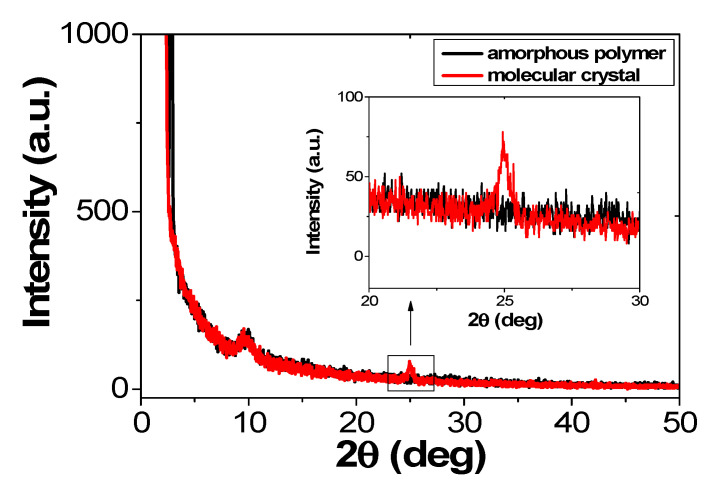
Measured XRD data of MEH-PPV film.

**Figure 7 molecules-26-02512-f007:**
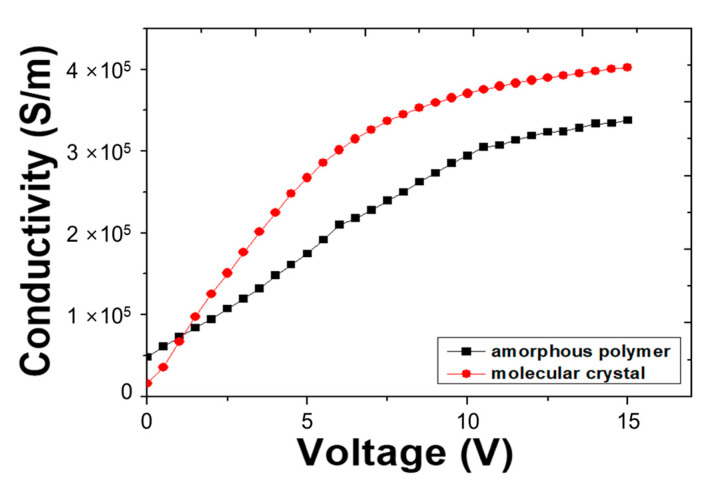
Conductivity of MEH-PPV film at 75 Hz by LCR meter.

**Figure 8 molecules-26-02512-f008:**
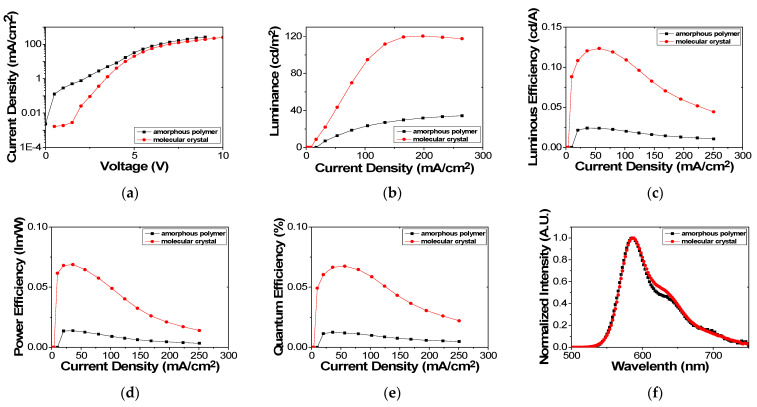
Electro-optical characteristics of OLED devices fabricated by spin coating and molecular ordering coating methods: (**a**) current density with log scale, (**b**) luminance, (**c**) luminous efficiency, (**d**) power efficiency, (**e**) quantum efficiency and (**f**) electroluminescence spectrum.

**Table 1 molecules-26-02512-t001:** The measured electro-optical characteristics of OLED device fabricated by amorphous polymer and molecular crystal.

Molecular State	V_on_	L_max_	LE (Max)	PE (Max)	EQE (Max)
amorphous polymer	5.0 V	35.75 cd/m^2^	0.024 cd/A	0.013 lm/W	0.012%
molecular crystal	4.5 V	120.3 cd/m^2^	0.123 cd/A	0.068 lm/W	0.066%

## Data Availability

Data is contained within the article.

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
