# Peer review of "Effects of MEH-PPV Molecular Ordering in the Emitting Layer on the Luminescence Efficiency of Organic Light-Emitting Diodes"

_molecules, 2021, doi:10.3390/molecules26092512_

Round 1

Reviewer 1 Report

The Paper “ Effects of MEH-PPV molecular ordering in the Emitting Layer on the Luminescence Efficiency of Organic Light-emitting Diodes” by Seok Je Lee et al. describes the effects of molecular ordering on the electro-optical characterisof OLEDs with an EML of MEH-PPV.

The concept is certainly neither new nor innovative, however an in-depth study on the structure-property relations in the emitting layers can always be interesting.

The fact that the efficiency in OLED devices can be influenced by the degree of order of the material because this directly affects the transport properties is known. Please add references

Heat treatments are usually used to improve the order of polymeric films but recently it has been shown that it is also possible through the use of an off-center spin-coating method and  many others see as an example Polymers 20179(6), 212; https://doi.org/10.3390/polym9060212. but there are many others that I suggest you consider and integrate in the paper.

For this reason it is necessary to review the introduction and the motivations of the work and to better focus the objectives. In addition, the references must be updated also taking into account the previous suggestions. The most recent dates back to 2018, then there are two references from 2015 and all the others are previous. Please update.

In the introduction it is important to give a definition/ explanation of the coating method that the authors intend to propose in comparison with the traditional spin coating. Please integrate.

I suggest adding a figure that contains the structure of the polymer used and the geometry of the device. It would also be useful to include the 2 different deposition methods in the figure.

Regarding  “ Results and discussion” it is necessary to give more information on the periodic pattern and in particular if the periodicity is compatible with the   the polymer structural features .

It is clear that the characteristics of a film change according to the substrate that is used or the layer directly in contact below . In the case of this work, glass for AFM, silicon wafers for SE, PEDOT for OLED measurements… were used. Did the authors take these differences into account in their data analysis? Please discuss.

In Fig. 1  an increase in the roughness of the base aluminum layer is evident passing from the spin coating to the molecular ordering method (see line 80-83). Do you think this can affect the parameters of the device? Adhesion to metal contact is usually a very important parameter in devices as demonstrated by the interfacial engineering obtained using conjugated polar polymers, the roughness can influence it? 

Is the periodic space mentioned at line140  compatible with the periodic order suggested? Please discuss and add a figure on the possible  polymer stacking in the plane; if necessary add molecular modelling.

The comparison of  the effect of the  “coating method” with the thermal annealing can provide important insights to understand the effect of the technique used. Please add it.

Line 156-159 “…..strong correlation between the molecular ordering and conductivity. Thus, the electrical characteristics can be improved by controlling the molecular ordering” please discuss and add new reference related to the material or similar polymers were is reported a correlation between order and conductivity

When comparing different devices, the main parameters that characterize them are usually shown in a table. Please add a table with this data by adding the EQE as well.

Line 163 “Finally, the electro-optical characteristics OLED with the molecular crystal and amorphous polymer were measured by the I-V-L system”. Are you sure it is a molecular crystal? Please discuss.

Luminescence and electroluminescence  data are missing. They are important in OLED characterization  and often the position and shape of the peaks are influenced by polymer packing. Please add and discuss appropriately.

At line 171-172 is reported  “The luminescence efficiency of the molecular crystal is 5 times higher than that of the amorphous polymer”. Is Luminescence or electroluminescence? Does the film obtained by spin coating have the same luminescence efficiency as the film obtained by coating? Are there any differences in these films ( spincoated and  "molecular ordering coating" films) between luminescence and electroluminescence?Please discuss and in the experimental EL and PLQY measurements.

Figure 6 (c) the luminous efficiency–current density just before 50mA increase suddenly is it an instrumental problem? Can you explain it?

Clearly the the charge injection strongly depends on the Homo Lumo levels of the materials used. It might be useful to add Levels or integrate them into a figure. Do you think that a change in morphology and order in the polymer can also affect energy levels? Please discuss.

The conclusions must be integrated with some consideration regarding the proposed technique, if it is competitive, if it can have future developments etc ....

More

What kind of solvent do you use for films preparation ?

Often the authors use the term molecule and polymer as synonyms, it is clearly not correct and therefore it is necessary to carefully check the manuscript

Reviewer 2 Report

The authors demonstrated the effects of molecular ordering on the electro-optical characteristics of OLEDs with an EML of MEH-PPV. The EML was fabricated by spin coating method. The luminous efficiency obtained with the polymer crystal was about 5 times higher than that of the amorphous polymer. This proposed strategy is interesting and effective, which is beneficial to the development of OLEDs. For these reasons, this paper can be accepted after solving below issues. 1. The authors are suggested to show the device structure of OLEDs in the Figure. 2. How about the power efficiency? 3. The authors are suggested to make some comments about how to further enhance the device performance. 4. Recent related papers about OLEDs should be cited (e.g., Adv. Funct. Mater. 2016, 26, 776-783; ACS Energy Lett. 2018, 3, 1531-1538; IEEE Electron Device Letters 2021, 42, 387)

Reviewer 3 Report

Authors of the manuscript „Effects of MEH-PPV molecular ordering in the Emitting Layer on the Luminescence Efficiency of Organic Light-emitting Diodes“ investigated the effects of molecular ordering on the electro-optical characteristics of organic light-emitting diodes.

 The manuscript would be interesting for researchers working in OLEDs field and could be published, but it needs some revision and explanations

  • Low efficiencies of the devices should be described in introduction and conclusions of the manuscript.
  • Brightness of the devices are very low. This should be explained in the manuscript.
  • In the introduction, there is described usually history of  OLEDs developments. It should be described also published research in the field of effects of molecular ordering for properties of OLED devices.
  • Electroluminescent spectra of the devices should be demonstrated. Do the different devices demonstrate the same spectra ?
  • Polymeric layers of the devices could be prepared by using different solvents. Could authors demonstrate the effect of solvent for the molecular ordering of materials and also for properties of the OLED devices?
  • Could the authors demonstrate efficient devices by using the described method ?

Round 2

Reviewer 1 Report

The revised manuscript : “ Effects of MEH-PPV molecular ordering in the Emitting Layer on the Luminescence Efficiency of Organic Light-emitting Diodes” has been strongly improved after the revision . The authors answered to the requests even if in my opinion they could include a greater number of recent references.

Anyway there are again some issue to be addressed.

 First of all I suggest to read carefully the whole manuscript because there are typos errors and the English is really poor and for this reason sometimes the meaning of the sentences is not clear.

Below together with the comments some of typos errors

Line 66 “are play” please correct

Line 77 “  proposed by khim et al. [24], the emission layer (EML)” please correct Khim and emitting

Line 79 “…ordering by two different” change in” ordering induced by….”

Line 80 “…caused by “ is better to use: due to.

Line 81 “…, and the electrical properties and the device…” and is a repeat please delete the first one

The paragraph “ Materials and Fabrication Methods” reports many considerations to be included in the results and discussion part. In general, the experimental and instrumental details are reported in this paragraph. The part relating to the discussion of the results should be moved to the correct paragraph.

I suggest to move lines 122-131; 136-141; 155-167.

Line 88 “…e hole transfer layer…”  It is hole transport layer; please correct.

Lines 88-90 “..The LUMO and HOMO energy levels …”  Regarding the Homo-LUMO levels of the materials : Have they been calculated experimentally or are you using values reported in the literature? add this detail and, if necessary, the necessary references.

Lines 91-92 “…substrates (15 Ω/□, active layer of 3 91 mm x 3 mm) substrates…” substrates is a repeat please delete one.

Lines 99-100 “….MEH-PPV  was mixed into…” please replace mixed with dissolved.

Line 103 “…. processes is shown …”  are shown…

Line 104 and Line 107  “…the solvent was dried…...” means anhydrification of the solvent. It is better to use : the solvent was removed or the film (the coating or the sample) was dried.

Lines 162-163 “… The LCR measurements were used to compare between the electrical conductivity 162 caused by the molecular ordering and amorphous polymer…”. ….. to make a comparison between…. or …..to compare the electrical…..

Caption Fig . 1  “…energy band diagram…” is better  “energy level diagram”.

Caption Fig. 2 “(c) mechanism of molecular alinment.”  Alignment? Please correct  

Line 189 “…The methods induce different orderings of the MEH-PPV molecules….” as suggested in my previous review, molecule and polymer are not synonymous and authors need to pay attention to the use of these terms. Molecules is not appropriate you can use macromolecules or better polymer chains

Lines 191-195 “… fabricated by the spin coating and molecular ordering coating methods…” this sentence is repeated 2 times in 3 lines . Please check.

Caption Fig 3 “…Figure 3. Ga-FIB-SEM images of MEH-PPV obtained by…” is better MEH-PPV film …..

Caption Fig 4 “….Figure 4. AFM images of MEH-PPV obtained by…” please add  “film” after MEH-PPV.

Table 2  The EQE is really low. Please discuss your result.

Line 426 “… molecular ordering of MEH-PPV in the EML o…” MEH-PPV film as EML

Lines 432-433 “..we analyzed the properties of the emission by using AFM, SE, XRD, and LCR meter…”

The sentence is not clear , do you analyse the emission with AFM? What about electroluminescence?

Reviewer 3 Report

The paper could be accepted after the revision.

Author Response

We are very thankful for your valuable comments and recommendation. Due to the warm and valuable comments, this manuscript has significantly been improved. I hope that our response is satisfactory to the reviewers and the manuscript now is publishable in your journal.